# Combustion and Emission Analysis of Spent Mushroom Compost and Forestry Woodchip for Management and Energy Production

Monica Carnevale [1], Enrico Paris [1,*], Beatrice Vincenti [1], Adriano Palma [1], Mariangela Salerno [1], Ettore Guerriero [2], Raffaele Mancini [3], Marco Calcopietro [3] and Francesco Gallucci [1,2]

1   Council for Agricultural Research and Economics (CREA), Center of Engineering and Agro-Food Processing, Via della Pascolare 16, 00015 Monterotondo, Italy
2   National Research Council of Italy, Institute of Atmospheric Pollution Research (CNR-IIA), Via Salaria km 29.300, 00015 Monterotondo, Italy
3   Aster Energetica Srl, Via Flaminia km 33.900, 00068 Rignano Flaminio, Italy
*   Correspondence: enrico.paris@crea.gov.it

**Abstract:** Forestry woodchip and spent mushroom compost have commercial potential as sustainable residues in biological and chemical processes for energy production. This study focuses on the evaluation of agri-food industry waste energy valorization, with the aim to reduce the valuable biomass utilization for energy production without decreasing the process quality, thereby pursuing economic and environmental advantages. Burning trials were conducted in a fluidized bed biomass plant provided with emission abatement systems. The biomass mixture used for combustion was composed of pine and oak woodchip and spent mushroom compost. The biomass used was first characterized through compositional and energetic analysis, and subsequently, during the burning tests, a monitoring sampling campaign was carried out to analyze the gas and particles emission. Optimal combustion conditions were observed during combustion, with good oxidation of the organic material, relatively high $CO_2$ production, and low CO concentration in flue gas. Nevertheless, $SO_2$ concentrations in the combustion flue gas are greater than those found in the combustion of the most commonly used biomasses. In fact, the mixture compositional characterization revealed a non-negligible concentration of sulfur, which explains the high values of $SO_2$ detected in emission. The obtained results confirm that controlled combustion, together with suitable biomasses utilization, preliminary characterization, and emission monitoring, are essential practices for the realization of a sustainable process, both from an energy and environmental point of view.

**Keywords:** waste management; energy production; circular economy; emission; agro-industrial biomass

## 1. Introduction

Biomass is one of the most relevant renewable fuels for clean energy production [1–5] and satisfies a requirement of about 15% of the world's energy consumption [6,7]. The climate change issue and international decarbonization objectives lead the scientific research to study and develop sustainable technologies for biofuel's valorisation. The European Commission has foreseen the achievement of the following objectives of carbon emission reduction: 20% by 2020, 40% by 2030, and 80–95% by 2050 [8]. The use of biomass for energy production through biochemical and thermochemical conversion processes increased in recent years because of their widespread availability, low environmental impacts, physic-chemical characteristics, and consolidated energetic exploitation operative conditions [9–12]. The nature of biofuel and combustion parameters influence the dynamic of the conversion process and the chemical composition of emission profiles [13]. Due to its environmental and human health impacts, the emission production phenomena from biomass combustion require apposite measurements, standards, and regulation methods [14,15] necessary for atmospheric pollutants monitoring and the assessment of plant

conditions [16,17]. The depletion of fossil fuel resources for economic activities, manufacturing, and agricultural industries moves the recent interest, in the agro-energy sector, to innovative and competitive biofuels as waste materials, beyond traditional biomass or dedicated to energy transformation, to be used in the conversion processes [18]. For example, the exploitation of organic waste and mixtures from the agri-food industry for energetic purposes, provides better management of residual materials, reducing the amount of waste and ensuring the green circularity of the production system thanks to their high conversion efficiency and low operative costs [19]. Recent studies about the combustion of solid biomass had the goal of improving plant efficiency and reducing pollutant emissions in the bioenergy field [20–23]. Biomass energetic exploitation represents a sustainable practice and contributes to producing clean energy, and yet, the relative emissions represent a notable environmental and health risk factor [24–26]. The aim of this work concerns the use of woodchip and spent mushroom compost (SMC) to produce a Mixture of Mushroom and Woodchip (MMW). Such a mixture was used for combustion in a fluidized bed plant (using olivine and sand as bed material), as an innovative solution to dispose of residual compost and forestry biomass through a thermochemical process able to produce energy. An MMW of pine and oak woodchips and SMC are abundant residual materials. Woodchips show several advantages, such as transport suitability, high energy density, and durability and are easily usable for heating systems and energy recovery [27]. Regarding SMC, over 6 million tons of edible mushrooms are produced worldwide annually [28], and over 30 million tons of mushroom compost waste are being produced [29]. Its disposal is necessary and foresees the shedding in the ground or landfill or its use as organic fertilizer and incineration [27,30,31]. Generally, the improper management of these latter waste products causes environmental pollution and requires high disposal costs. For this reason, the energy valorisation of such combined waste materials by thermal conversion process represents an innovative strategy for the sustainable management of SMC residues, with an energetic potential comparable to other waste, such as wood biomass and municipal solid waste [32–37]. Firstly, the matrices were chemically and physically characterized; subsequently, the mix was converted by combustion in a fluidized bed boiler. The mixture of woodchip and compost resulted in a potential renewable biofuel, and its application in systems with proper uses and sizes for clean energy production [38,39] can contribute to mitigating environmental impacts, reducing greenhouse gas emissions, air pollutants, and improving the waste management policies towards the circular economy [30,32].

## 2. Materials and Methods

Placed in the context of residual materials energy valorisation, the present work has allowed the optimization of the thermochemical conversion process of SMC mixed with 30% wood chips (pine 50% and oak 50%) and highlighted critical issues that will require further optimization. An analysis of bed combustion, before and after the conversion process, was conducted together with the produced ash analysis and incoming biomass chemical-physical characterization. During combustion, an accurate emissions monitoring campaign was carried out, analysing macro-pollutants, particulates, and metals The power plant operated in a continuous mode, regulating the power output as a request by the user; plant switch off was realized for maintenance at scheduled intervals. The test duration lasted about three hours, during which the plant power output was been maintained as much as possible in a steady way.

### 2.1. Biomass Characterization

In the LASER-B (Laboratory of Experimental Renewable Energies from Biomass) of CREA-IT, the physical and chemical analyses were conducted. The biomass was supplied by a historical mushroom farm located in Lazio, Italy, where the biomass test plant realized by Aster Energetica Srl is also located. The moisture content was measured by means of a Memmert UFP800 oven, according to the UNI EN ISO 18134-1: 2015. In particular, the biomass sample was dried at $105 \pm 2$ °C, and the moisture content was determined

considering the weight loss of the sample due to the drying process after 24 h. The ash content was determined using a Lenton EF11/8B muffle furnace according to the UNI EN ISO 18122: 2016. The sample of about 1 g was placed in a ceramic crucible and then inserted into the furnace, according to the thermal ramp; specifically, the oven temperature was set with two different temperature rates, the first step at 6 °C/min up to 250 °C and the second at 10 °C/min from 250 °C to 550 °C. The higher heating value (HHV) was determined by a Paar 6400 calorimeter, according to the UNI EN ISO 18125: 2018. The lower heating value (LHV) was determined from the HHV considering the hydrogen content. The elemental composition was determined using a Costech ECS 4010 CHNS-O elemental analyser and according to the UNI EN ISO 16948:2015. As regards the determination of volatile content and fixed carbon, the UNI EN 13649:2015 standard method was used as reported by Carnevale et al. [40]

### 2.2. Experimental Combustion Test and Emission Monitoring

The combustion test was conducted in a fluidized bed boiler (model Aster CLF500: 500kW thermal power output); the technology is based on the fluidized bed principle, which is a sand bed with a suitable particle size maintained in a fluid state by air injection. The combustion process is preliminarily performed into the bed, which operates at a relatively low temperature of <650 °C. Biomass has a high moisture content, which inevitably reduces the bed temperature and the efficiency of the thermochemical conversion process. Although the temperature is lower than other similar works in the literature [41], it was observed during the preliminary stages of study in the process that this temperature is optimized for this type of biomass.

Biomass volatile fraction is separated from the fixed fraction into the bed, and this volatile gas is subsequently oxidized in two combustion chambers (an accelerating chamber and a stationary chamber) at an operative temperature of 850–900 °C. The Aster CLF boiler represents an evolution of this technology, applied to the biomass sector (Figure 1). The combustion of biomass inside the fluidized bed is visually characterized by the reduced presence of flames. The boiler is designed and developed to ensure high flexibility with the use of numerous types of biofuels with low energy content, even when mixed, always guaranteeing perfect combustion and reducing polluting emissions. In traditional biomass combustion systems (fixed grate, mobile grate), due to the complications deriving from the use of biomass with high moisture and high ash content, there is usually frequent and unwanted downtime; this is mainly due to the fouling of the heat exchanger surfaces and the accumulation of agglomerates by melting the aggregates at relatively low temperatures, effectively preventing regular combustion and emissions within the limits. The boiler used in this work allows the possibility to add additives directly into the combustion chamber, the dry dedusting and the particulate matter (PM) abatement by multi-cyclone filter and bag filter in series. The CLF boiler's main technical data are shown in Table 1.

**Table 1.** Boiler data at combustion test conditions.

| Boiler Data and Operative Parameters | Min | Max | Unit |
|---|---|---|---|
| Thermal power produced | 310 | 330 | [kW$_{th}$] |
| Fuel flow | 295 | 320 | [kg/h] |
| Primary air flow | 680 | 720 | [Nm$^3$/h] |
| Secondary air flow | 325 | 850 | [Nm$^3$/h] |
| Gas flow | 1850 | 2000 | [Nm$^3$/h] |
| Exchanger inlet fumes temperature | 825 | 858 | [°C] |
| Boiler outlet fumes temperature | 155 | 158 | [°C] |

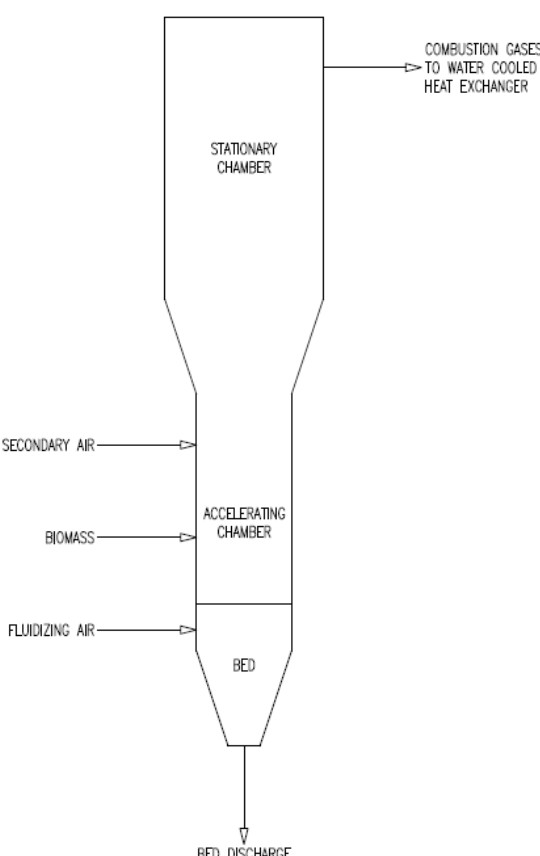

**Figure 1.** The heat exchanger was positioned downstream the oxidating chamber and produced approximately 500 kWt. Thermal power is provided to the user through the superheating of the water. The two combustion chambers (accelerating and stationary flue gas) are connected as represented.

The analysis of gaseous macro-pollutants from the stack was carried out using an HP1 DADOLAB heated sampling probe. The probe was connected to two heated lines (180 °C), which convey the gaseous effluents from the sampling point to the analysers. Two in-line gas analysers were used: a Ratfisch RS 53-T (FID) heated flame ionization detector for TOC (Total Organic Carbon) and a multiparametric analyser (Horiba PG250) for $NO_x$, $SO_2$, CO, $CO_2$, and $O_2$ monitoring. A probe for isokinetic sampling (DADOLAB HP5 + ST5) was interfaced with the stack for the determination of the fume's moisture, Total Suspended Particles (TSP), and metals. The probe housed a quartz wool filter (∅ 47mm) in a heated chamber for the PM deposition. Such filters were conditioned at 400 °C and subsequently brought to a constant weight by keeping them for 48 h at a constant temperature and moisture in a conditioning chamber. After sampling, filters were weighed in order to derive the TSP for gravimetry. Filters were then mineralized using a solution of $HNO_3$ and $H_2O_2$ in a Milestone StarD mineralizer according to EPA Method 3051A. The sampling of metals involves, downstream of the filter, the use of a bubblers system placed in line in a refrigerated chiller at 5.0 ± 0.1 °C, in accordance with UNI EN 14385:2004. Filters allowed to trap the PM coarse fraction while the bubblers allowed the condensation and trapping of the volatile metals. The three bubblers were filled with 100 ml of a MilliQ water solution containing $HNO_3$ (3.25% *v/v*) and $H_2O_2$ (1.5% *v/v*). The third bubbler is defined as a "backup" and validates the sampling; in this solution, the concentration of the metals must be less than 10% compared with the sum of the first two bubblers. The mineralized filter and the bubbled solutions were analysed in the Agilent 7000 ICP-MS for the determination of the metallic species present according to UNI EN ISO 16967:2015. The monitoring campaign also concerned the sampling of PM fractions, i.e., $PM_{10}$, $PM_{2.5}$, and the intermediate $PM_{10-2.5}$ fraction. This latter sampling was carried out

using a new inertial impactor prototype developed by DADOLAB Srl in compliance with ISO/CD 23210. The data relating to the stack sampling conditions are reported in Table 2.

**Table 2.** Stack sampling parameters and operative conditions.

| Parameter | Unit | Value |
|---|---|---|
| Diameter | [m] | 0.35 |
| Area | [m$^2$] | 0.09 |
| Density | [kg/Nm$^3$] | 1.31 |
| Moisture | [%] | 12 |
| Velocity | [m/sec] | 13.60 |
| Stack temperature | [°C] | 138.66 |
| Stack pressure | [kPa] | 99.38 |
| Velocity at nozzle | [m/sec] | 13.52 |
| Prope temperature | [°C] | 134.5 |
| Filter temperature | [°C] | 120.3 |
| Environment pressure | [kPa] | 99.39 |

### 2.3. Sampling and Characterization of Combustion Bed and Ashes

The study also concerned the characterization of the combustion bed material and the ashes produced during the thermochemical process. In particular, the bed material used (olivine and sand) was characterized before and after the combustion process to evaluate any chemical alterations suffered by the material because of the thermal conversion. According to a previous study [42] the bed material can lead to overestimating the metals produced by biomass combustion. During the process, bed material (composed of olivine/sand and bottom ashes) was extracted continuously by means of a discharge system, which provided the separation of the bottom ashes from the olivine/sand and the re-feeding of the olivine/sand into the bed. Sampling was conducted by extracting bed material samples from the discharge system before the re-feeding phase.

The olivine/sand mixture was characterized by a thermogravimetric analyzer (TGA-DSC 1 STAR System, Mettler Toledo), to evaluate the changes in weight and heat exchange during the temperature increase. About 11 mg of material were placed in the crucible of the TGA-DSC with air as carrier gas (25 mL/min) and a temperature rate of 80 °C/min from 25 °C to 850 °C. The ashes and the bed material were also mineralized to evaluate the metal content in ICP-MS. The residual ash was also investigated by SEM Zeiss EVO MA 10 + EDS Bruker Quantax 200 to evaluate the morphology and surface chemical composition.

### 3. Results and Discussion

### 3.1. Biomass Characterization

The chemical and physical composition of biomass is related to the type of species used, the soil on which it grows and strongly influences the ash production, and pollutants emission during combustion. The high content of ash represents one of the most important issues of biomass waste combustion, in particular for the bed agglomeration effect, slagging, fouling and corrosion [1,43,44], due to some elements like Si, K, Na, S, Cl, P, Ca, Mg, Fe. Therefore, before each test, it is necessary to investigate the chemical and physical parameters of the matrix used in the plant to prevent damage to the system. Combustion tests were conducted by using MMW composed of 70% from SMC (a mixture of straw, chicken litter and gypsum) and 30% from pine and oak woodchips (1:1). This percentage ratio is a result of a process of optimization of the incoming biomass studied so that it was possible to use the highest possible percentage of SMC, the main object of investigation of the research conducted. 30% of lignocellulosic biomass was however necessary to allow a good combustion process (in terms of temperature, turbulence, and the combustion gas's time residence) and give better structural properties to the fuel. For completeness, both pure SMC and mixture characterization data were reported. It is important to underline that compost is a highly volatile matter biomass given the presence of straw. Consequently,

the bulk of the combustion takes place in the downstream chambers and obviously before the exchangers where all the volatile part is oxidized. The wood chips' behaviour is instead very different; in fact, the fixed carbon is higher, and this leads to a high temperature in the bed/first chamber and lowers in the subsequent chambers. The obtained results from the analysis of the samples are reported below (Table 3). With respect to the use of conventional lignocellulosic biomass, the SMC shows higher N and S content, higher moisture percentage and lower carbon content. Unavoidably, such characteristics strongly influence the mixture composition and energetic behaviour.

**Table 3.** Proximate and ultimate analysis of SMC and its mixture with woodchips MMW.

| Biomass | Ash [%] | Moisture [%] | Fixed carbon [%] | Volatile [%] | C [%] | H [%] | N [%] | S [%] | HHV [MJ/kg] | LHV [MJ/kg] |
|---|---|---|---|---|---|---|---|---|---|---|
| MMW | 19.71 | 51.13 | 3.16 | 26.00 | 31.97 | 2.18 | 2.01 | 1.29 | 17.11 | 16.66 |
| SMC | 12.10 | 65.55 | 2.15 | 20.20 | 11.0 | 1.1 | 1.7 | 1.81 | 17.04 | 15.25 |

Metal Content

Result of metal concentration are reported in Table 4. A significant amount of K and Ca can be found in the agricultural practices by fertilizers [45]. These abundant elements mostly cause phenomena as fouling and slagging, corrosion, and particulate emissions [13,46], which follows the formation of silicates, carbonates, sulphates, and some oxyhydroxides, phosphates and nitrates [47].

**Table 4.** Metal content of SCM and MMW.

| [mg/kg] | MMW | SMC |
|---|---|---|
| Li | 6.63 | 3.78 |
| B | 25.96 | 20.62 |
| Na | 2249.85 | 3247.60 |
| Mg | 5507.92 | 5270.18 |
| Al | 5050.91 | 4127.11 |
| K | 21,222.47 | 29,463.62 |
| Ca | 5424.93 | 45,854.00 |
| Cr | 5.85 | 2.57 |
| Mn | 227.43 | 238.17 |
| Fe | 4187.94 | 3962.32 |
| Co | 1.94 | 2.01 |
| Ni | 5.95 | 9.85 |
| Cu | 30.37 | 33.95 |
| Zn | 79.02 | 165.81 |
| Ga | 10.36 | 8.55 |
| Sr | 269.33 | 178.21 |
| Ag | <LOQ | <LOQ |
| Cd | 0.09 | 0.38 |
| In | <LOQ | <LOQ |
| Ba | 161.03 | 100.61 |
| Tl | 0.36 | <LOQ |
| Pb | 5.02 | 2.12 |
| Bi | <LOQ | <LOQ |

*3.2. Characterization of Fluidized Bed*

3.2.1. Thermogravimetric Analysis

The thermogravimetric analysis was carried out to measure the sample mass loss as the temperature increases under controlled conditions [3,48–50], simulating the thermo-chemical processes dynamic [51,52].

The use of olivine as bed material, in partial or total replacement of silica sand, is related to the consistent reduction of the agglomerates formation due to the localized fusion of low-melting ash present in the treated biomasses. Olivine was studied in TGA-DSC to evaluate the behaviour of the material during the combustion process. The results of olivine thermal behaviour (weight loss and heat flow trend) are shown in Figure 2.

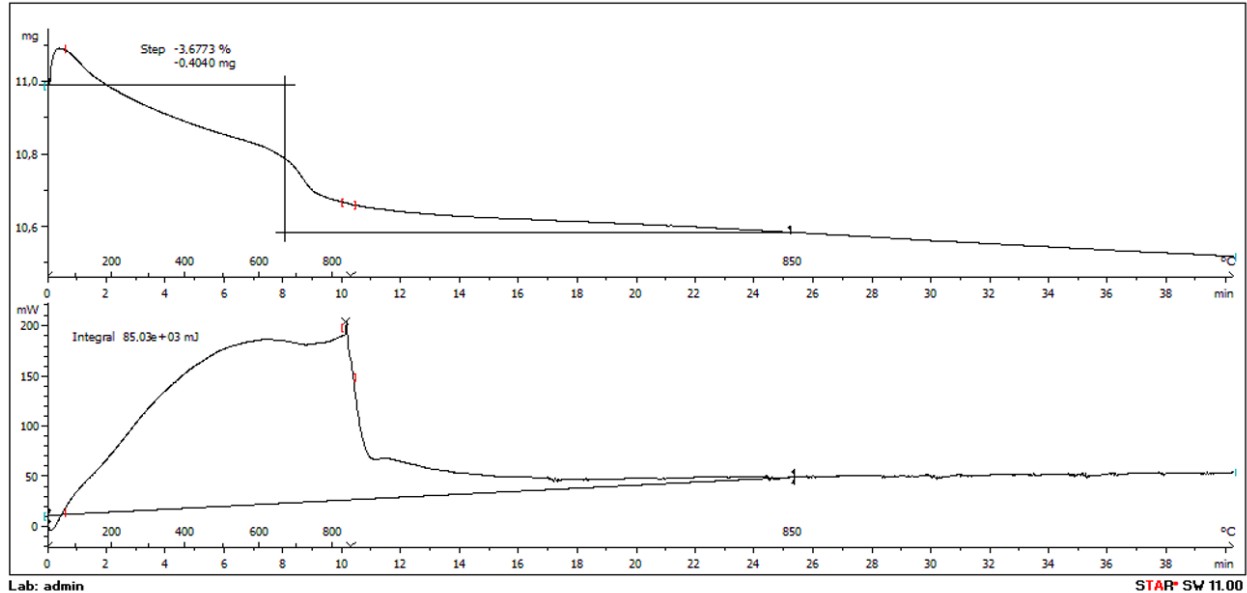

**Figure 2.** TGA-DSC Olivine/sand analysis.

In the range of 25–850 °C, the material has a weight loss of 4%, while as regards the heat flow graph it has a single exothermic peak. The olivine thermal behaviour is in line with the results shown in Table 5, where at the end of the combustion process some of the metals in the emission are attributable to the bed material.

**Table 5.** Metal content in olivine before and after combustion process.

| [mg/kg] | Olivine/Sand | Bed Material | Δ |
|---|---|---|---|
| K | 23.05 | 13,752.25 | 13,729.20 |
| Ca | 325.00 | 5669.26 | 5344.26 |
| Al | 1948.87 | 5515.90 | 3567.02 |
| Na | 102.92 | 1249.87 | 1146.95 |
| Sr | 1.92 | 189.02 | 187.11 |
| Zn | 7.91 | 129.30 | 121.38 |
| Ba | 0.67 | 63.18 | 62.51 |
| B | <LOQ | 52.63 | 52.63 |
| Mn | 582.41 | 606.74 | 24.32 |
| Li | 1.14 | 7.07 | 5.93 |
| Cu | 13.14 | 16.80 | 3.66 |
| Ga | 0.22 | 2.90 | 2.68 |
| Pb | 0.04 | 1.25 | 1.21 |
| Co | 67.59 | 50.09 | −17.51 |
| Cr | 183.64 | 121.02 | −62.62 |
| Ni | 1292.55 | 941.62 | −350.93 |
| Fe | 36,207.20 | 24,281.19 | −11,926.01 |
| Mg | 162,477.03 | 120,670.85 | −41,806.18 |
| Ag | <LOQ | <LOQ | <LOQ |
| Cd | <LOQ | <LOQ | <LOQ |
| In | <LOQ | <LOQ | <LOQ |
| Tl | <LOQ | <LOQ | <LOQ |
| Bi | <LOQ | <LOQ | <LOQ |

### 3.2.2. Metal Content

As mentioned in Section 3.1, the high content of ash may limit biomass use because it adversely affects the combustion conditions. In addition, a high proportion of ash may adversely affect the metal's emission. For this reason, several studies evaluate the optimal combination of bed materials [53,54] to minimize the operating problems related to metals [43,45,55]. The fluidized bed boiler doesn't collect separately, determine, or characterize the residual ashes in the combustion chamber (bottom ash). Therefore, in order to characterize the presence of metals in the bottom ash, it was chosen to analyze the material of the bed before and after the combustion process to obtain the delta that can express reliable data on the bottom ash characterization. In Table 5, it is possible to compare the content of metals in olivine before and after the biomass combustion process.

Observing the Δ of concentration in Table 5, there is an enrichment of metals in the bed after the combustion process. This is mainly due to the presence of ashes. In fact, at the end of the process, the biomass ashes are mixed and aggregated with the bed material. However, in the case of cobalt, chromium, nickel, iron, and magnesium, the delta is negative. This implies that these metals are most likely released from the bed material and their presence in the emissions will be due to the thermal stress to which the bed is subjected and not exclusively to the biomass. Iron and magnesium are the main metals released since they are the main constituents of olivine [$(Mg, Fe)_2SiO_4$]. Other metals are probably due to impurities naturally present in the mineral.

### 3.3. Characterization of Emissions

Biomass burning is responsible for atmospheric emission generation and great inorganic residues production, distributed between the gaseous and solid fraction [56]. Gas emission concentrations obtained during the six hours of sampling are shown in Table 6, and relative trend emission flow is shown in Figures 3–6. The results expressed in $mg/Nm^3$ are compared with an oxygen content of 11% in compliance with the Legislative Decree 183/17.

**Table 6.** Conversion into $mg/Nm^3$ and related to 11% of $O_2$.

| CO [$mg/Nm^3$] | NO$_x$ [$mg/Nm^3$] | SO$_2$ [$mg/Nm^3$] | TOC [$mgC/Nm^3$] |
|---|---|---|---|
| 4.94 | 298.63 | 263.45 | 1.55 |
| | 11% O$_2$ | | |
| 8.18 | 494.42 | 436.18 | 2.57 |

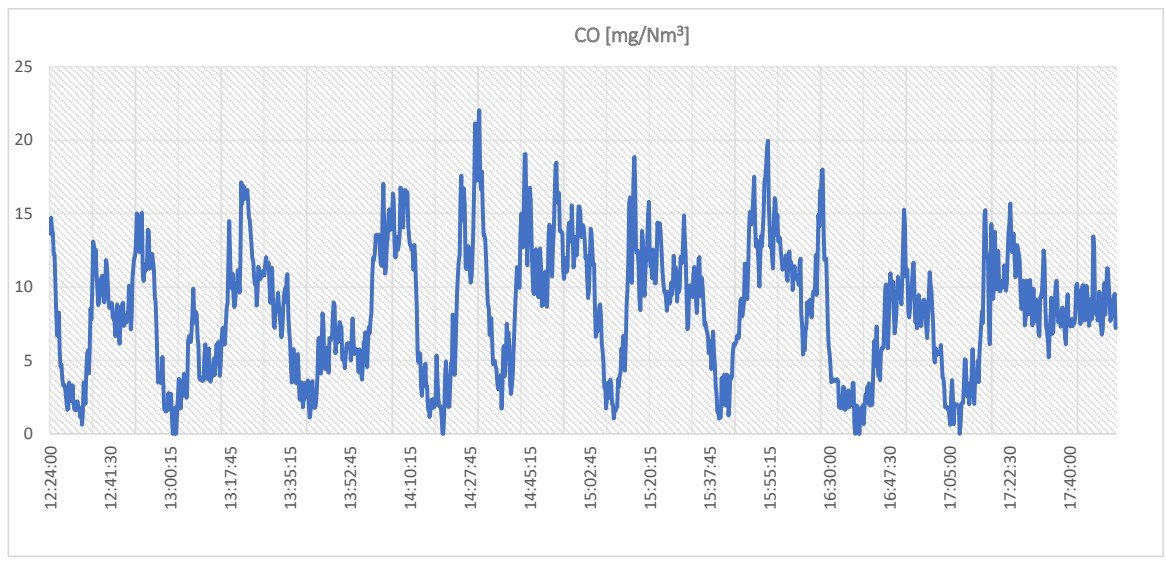

**Figure 3.** CO emission trend during combustion test.

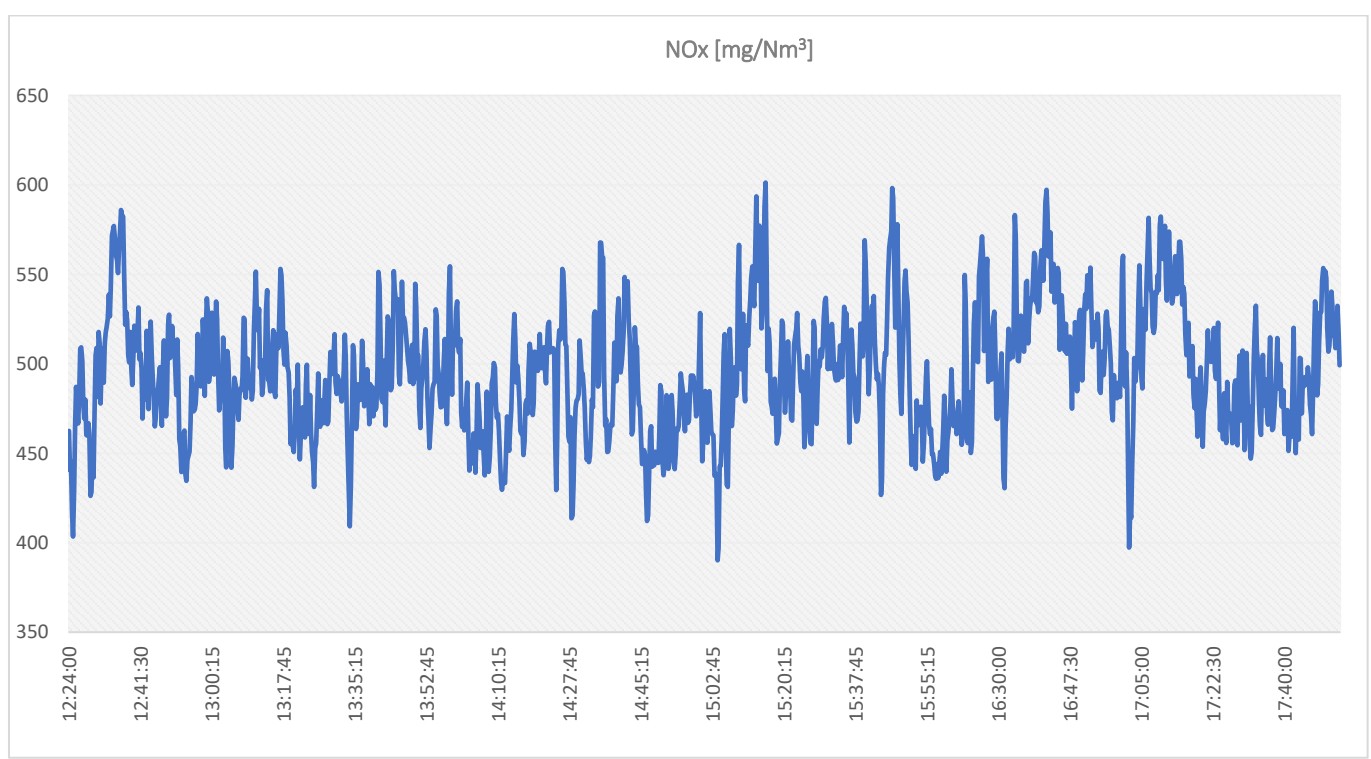

**Figure 4.** NOx emission trend during combustion test.

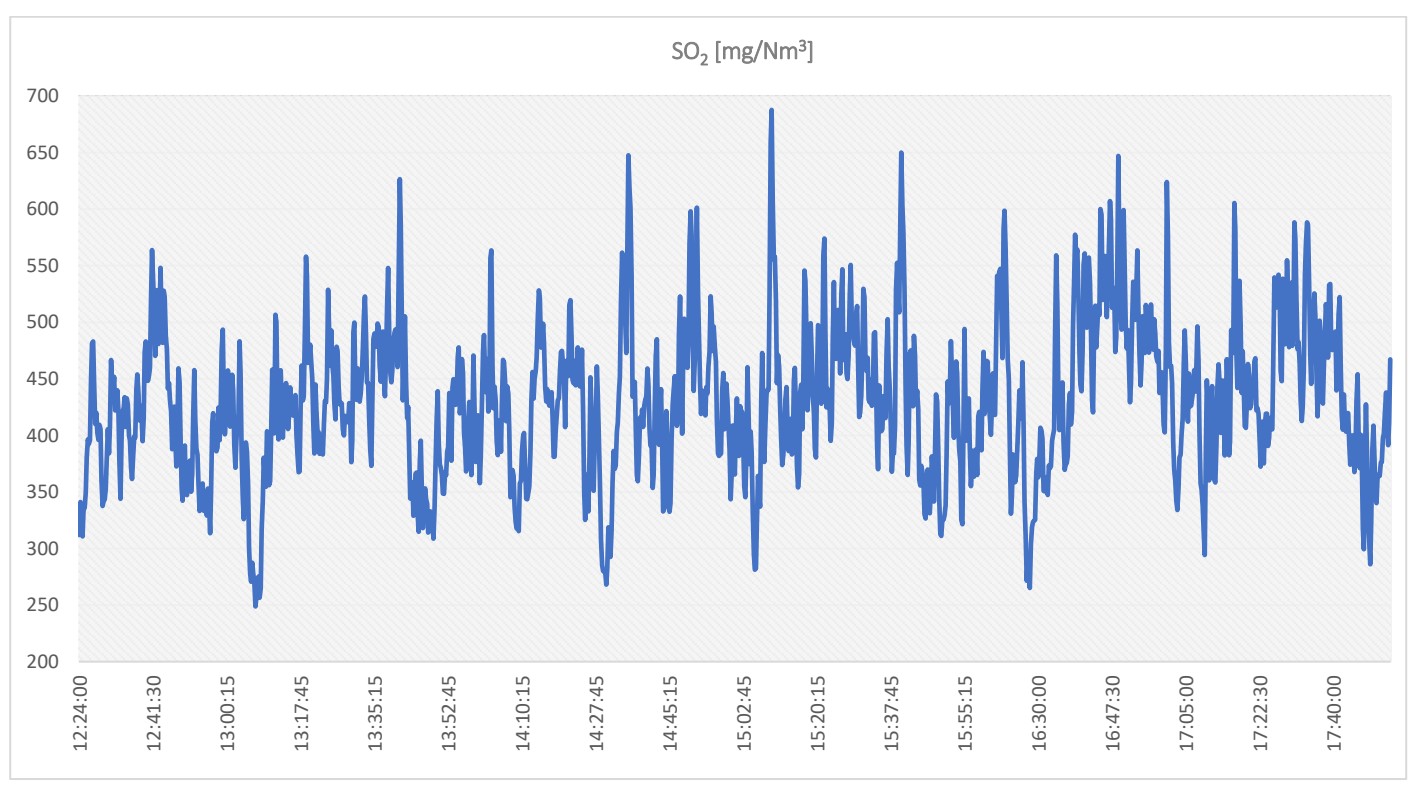

**Figure 5.** SO$_2$ emission trend during combustion test.

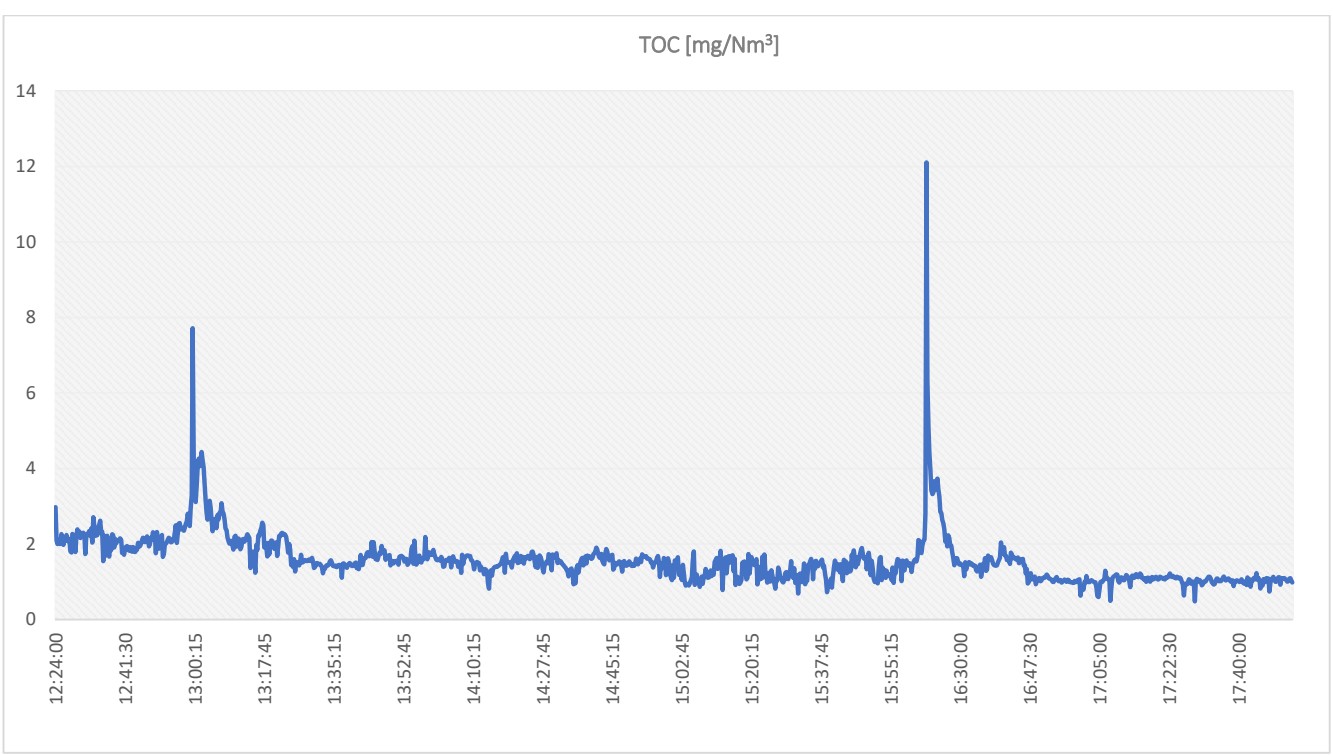

**Figure 6.** TOC emission trend during combustion test.

Results show good combustion conditions in terms of atmospheric emissions production. In particular, low values of carbon monoxide indicate combustion in good stoichiometric conditions, and this, therefore, suggests that there is a low profile of organic pollutants in emission (which are produced in large quantities in conditions of partial and/or not complete combustion of biomass). Comparing the values obtained with previous works in which other types of biomasses were tested in smaller boilers (spent coffee ground in an 80 $kW_{th}$ [57] and citrus, olive, and grapevine in a 30 $kW_{th}$ boiler, [58], we observe a lower concentration of $NO_x$ in emission, but a higher concentration of $SO_2$. This is because the SMC is particularly rich in sulphur.

It is estimated that almost 70% of $PM_{2.5}$ comes from wood burning in fireplaces and 30% from wood boilers and depending on the season, $PM_{2.5}$ emissions range from 30 to 90% and $PM_{10}$ from 10 to 80% [59]. The gravimetric analysis of the filters obtained the results for the TSP and the PM fractions. The results are reported as measured values normalized refer to 11% of $O_2$ (Table 7).

**Table 7.** Total Suspended Particles and PM fractions.

| $PM_{2.5}$ [mg/Nm$^3$] | $PM_{2.5-10}$ [mg/Nm$^3$] | $PM_{10}$ [mg/Nm$^3$] | TSP [mg/Nm$^3$] |
|---|---|---|---|
| 31.51 | 6.67 | 2.61 | 83.92 |
| | **11% O$_2$** | | |
| 52.17 | 11.05 | 4.33 | 138.71 |

The bag filter effect is evident. This abatement system effectively retains the PM generated with better results for larger fractions. The TSP are slightly above the legislative limits, and this is probably due to the high ash content of the incoming matrix. Figure 7 shows SEM image of the sampled PM. There are different morphologies and amorphous structures that incorporate more defined crystalline structures.

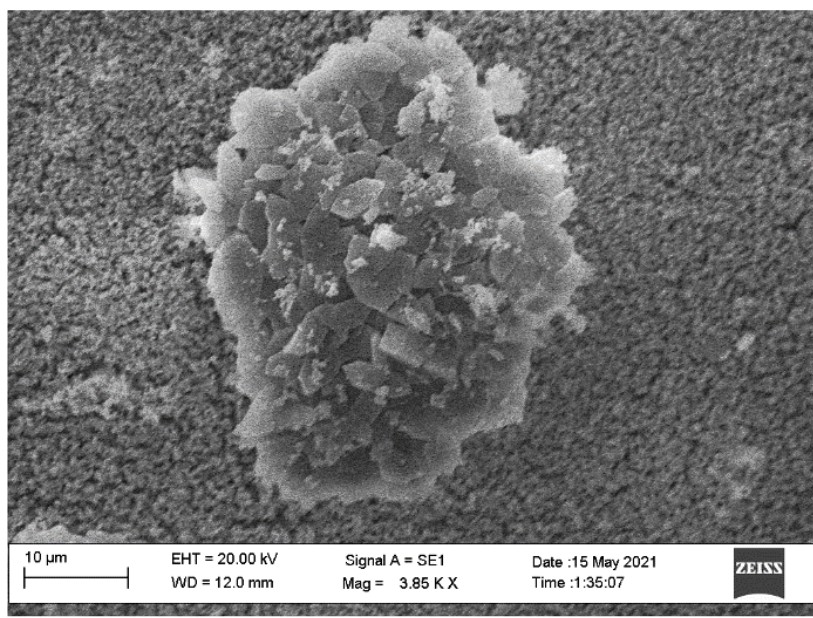

**Figure 7.** SEM image of PM on filters.

### 3.3.1. Metals in Emission

Result of metal concentration in emissions are reported in Table 8. In general, high concentrations of metals are not observed. The main metals emitted are macro-elements, due to the high concentrations of ash within biomass. Mg and Fe considerable concentration in emissions are due in part to the olivine of the bed.

**Table 8.** Distribution of metals emission between filter and volatile phase (bubblers).

| [mg/Nm$^3$] | Filter | Bubblers | Total |
|---|---|---|---|
| Li | 0.0028 | <LOQ | 0.0028 |
| B | 0.1793 | <LOQ | 0.1793 |
| Na | 1.2805 | 0.1517 | 1.4322 |
| Mg | 1.4418 | 0.0620 | 1.5039 |
| Al | 1.8454 | 0.0560 | 1.9015 |
| K | 1.5716 | 0.1652 | 1.7369 |
| Ca | 1.3729 | 0.0778 | 1.4507 |
| Cr | 0.0043 | 0.0006 | 0.0049 |
| Mn | 0.0608 | 0.0018 | 0.0626 |
| Fe | 1.8324 | 0.0521 | 1.8845 |
| Co | 0.0008 | <LOQ | 0.0008 |
| Ni | 0.0044 | 0.0016 | 0.0060 |
| Cu | 0.0176 | 0.0052 | 0.0228 |
| Zn | 0.0590 | 0.0460 | 0.1050 |
| Ga | 0.0021 | 0.0006 | 0.0028 |
| Sr | 0.0698 | 0.0019 | 0.0717 |
| Ag | <LOQ | <LOQ | <LOQ |
| Cd | 0.0005 | 0.0037 | 0.0041 |
| In | <LOQ | <LOQ | <LOQ |
| Ba | 0.0314 | 0.0123 | 0.0437 |
| Tl | 0.0015 | 0.0002 | 0.0017 |
| Pb | 0.0140 | 0.0204 | 0.0344 |
| Bi | <LOQ | <LOQ | <LOQ |

### 3.3.2. Metals in Ash

Especially important for the boiler performance is the role of the major and minor elements of ashes. Major elements such as: Ca, K, Mg, Mn, and P [60–63] have relevance in the ash melting behavior, slagging, deposits formation and corrosion problems. Minor elements such as As, Cd, Ba, Cr, Cu, Ni, Pb, and Zn [64–67] affect the toxicity of particulate emissions and prevent the possibility of ash utilization [68].

SEM-EDS analyses were conducted in order to observe the surface composition of residual ash. Unlike the analyses reported in Table 9 that show the composition of the total ashes, in Figure 8 and Table 10, only the surface layer is analyzed, and a series of elements of fundamental ash constitution (Si, S, P) can be detected.

**Table 9.** Metals content of ash after combustion.

| [mg/Nm$^3$] | Ash |
|---|---|
| Li | 23.66 |
| B | 70.89 |
| Na | 6031.09 |
| Mg | 14,103.48 |
| Al | 89,890.02 |
| K | 51,842.23 |
| Ca | 13,436.21 |
| Cr | 24.72 |
| Mn | 541.68 |
| Fe | 11,982.08 |
| Co | 5.01 |
| Ni | 16.34 |
| Cu | 68.77 |
| Zn | 156.91 |
| Ga | 26.17 |
| Sr | 700.02 |
| Ag | <LOQ |
| Cd | 0.10 |
| In | <LOQ |
| Ba | 375.88 |
| Tl | 0.69 |
| Pb | 13.22 |
| Bi | <LOQ |

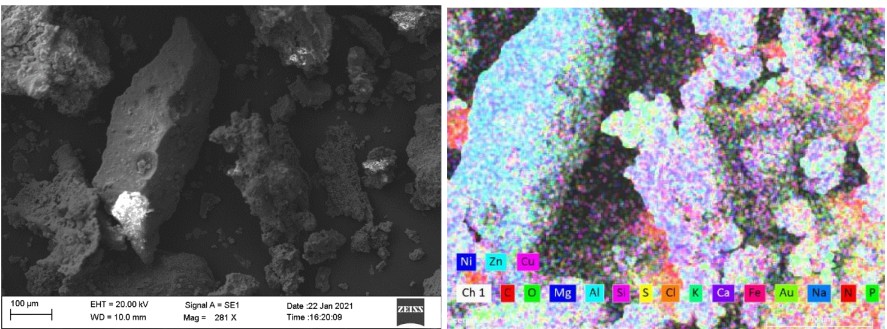

**Figure 8.** SEM image of ash (**left**), SEM-EDS scan and surface composition (**right**).

**Table 10.** Surface composition of ash by means SEM-EDS.

| O [%] | C [%] | Ca [%] | Si [%] | K [%] | S [%] | Al [%] | Na [%] | Mg [%] | Fe [%] | Zn [%] | Cl [%] | P [%] | Cu [%] |
|---|---|---|---|---|---|---|---|---|---|---|---|---|---|
| 23.9 | 18.5 | 11.4 | 7.96 | 6.10 | 2.40 | 1.67 | 1.68 | 1.24 | 1.08 | 1.06 | 0.62 | 0.57 | 0.21 |

### 3.3.3. Emission Factors

From the interpolation of the data reported in Table 8 and the parameters monitored during the combustion process, the emission factors were calculated. The results are shown in Table 11. As can be deduced from Table 5, these data are inevitably affected by the bed material emissions, and, therefore, it is likely that Co, Cr, Ni, Fe and Mg emission factors may have lower values in other combustion conditions, such as in a fixed bed boiler or different bed materials.

**Table 11.** Emission factors.

| EF | [mg emitted/kg fuel] |
|---|---|
| Li | 0.027 |
| B | 1.704 |
| Na | 13.613 |
| Mg | 14.294 |
| Al | 18.073 |
| K | 16.509 |
| Ca | 13.789 |
| Cr | 0.047 |
| Mn | 0.595 |
| Fe | 17.912 |
| Co | 0.008 |
| Ni | 0.057 |
| Cu | 0.217 |
| Zn | 0.998 |
| Ga | 0.027 |
| Sr | 0.681 |
| Cd | 0.039 |
| Ba | 0.415 |
| Pb | 0.327 |

The graph in Figure 9 highlights a trend of metals and some elements seem to share the same behaviour following a trend that could define them as more or less volatile. For example, from the results about metal content in the previous tables, the emissions have high values of Na, Al, K, Mg, and Ca, which come from the biomass, while Fe, Ni, Cr and Cd come from the combustion bed and are found in the ashes. Therefore, we can deduce that some elements by their nature tend to go into emission, while those tend to go into ashes. Observing the profile of the graph in its entirety produced by the union of points, it is observed that some elements (Li, Cr, Co, and Ga) have low emission concentrations with respect to their ash content and in the input biomass. In fact, in the emission profile (blue line Figure 9) there are more marked concavities for these elements.

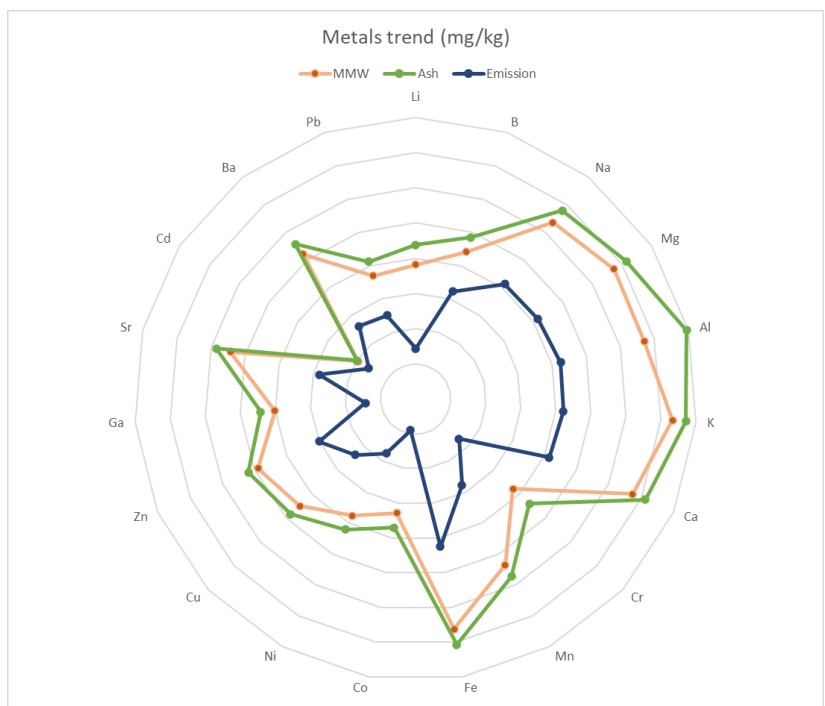

**Figure 9.** Metal trend in MMW, ash and emission.

## 4. Conclusions

The proposed work has evaluated the use of residual biomass waste in the food chain for sustainable disposal, which allows energy production. In particular, it was observed that mixing this matrix with waste from the forest (wood chips) has obtained a good fuel used in a fluid bed boiler fed with biomass. The main focus of the research was on the assessment of air emissions. During the emission sampling, low $O_2$ values and low CO values were recorded, this is an indication of good combustion conditions. The high values of PM and $SO_2$ can instead probably be attributed to the nature of the incoming biomass that shows, sulphur concentration and ash percentage higher than common lignocellulosic biomass The metals in emission are mainly made up of macro-elements (Ca, Mg, K, etc.) as usually occurs in the case of biomass combustion. As for the bed combustion material, it was shown that the use of olivine leads to the emission of some metals and minerals. Specifically, it has been seen how the comparison between the concentrations in the olivine analysed before and after the combustion process, shows negative deltas in the cases of Fe, Mg, Ni, Cr, Co. This result shows that the determination of the emission metals, generated by the combustion of biomass, can be influenced by the effect of the bed material and other internal boiler metal components (e.g., heat exchanger). Results suggest the possibility to lead further studies at different operative conditions (thermal power output, biomass mixing, bed composition) and monitoring systems, in order to reduce emissions, especially for $SO_2$ and $PM_{2.5}$.

**Author Contributions:** Conceptualization, R.M., M.C. (Marco Calcopietro) and F.G.; methodology, F.G., R.M., M.C. and E.G.; validation, E.G., E.P. and F.G.; formal analysis, M.C. (Monica Carnevale), B.V., E.G., A.P. and E.P.; investigation, M.S., M.C. (Monica Carnevale) and R.M.; resources, M.C. (Marco Calcopietro) and R.M.; data curation, E.P., E.G., A.P., B.V. and M.C. (Monica Carnevale); writing original draft preparation, M.C. (Monica Carnevale), E.P., A.P. and B.V.; writing review and editing, E.P., A.P., B.V., M.C. (Monica Carnevale) and R.M.; visualization, M.S., R.M. and M.C. (Marco Calcopietro); supervision, E.G. and F.G.; project administration, F.G.; funding acquisition, R.M., M.C. (Marco Calcopietro) and F.G. All authors have read and agreed to the published version of the manuscript.

**Funding:** This research was funded by the Regione Lazio—Lazio Innova spa, POR FESR 2014/2020 Regione Lazio—Avviso Pubblico "Circular Economy e Energia"—Programma Operativo Regionale 2014-2020, under the Project COMPOSTEAM—Riuso e valorizzazione degli scarti organici nelle aziende agro-alimentari da impiegare per la produzione di calore. This work was supported by the AGROENER (D.D. n. 26329, 01/04/2016) and sub-project "Tecnologie digitali integrate per il rafforzamento sostenibile di produzioni e trasformazioni agroalimentari (AgroFiliere)" (AgriDigit programme), (DM 36503.7305.2018 of 20/12/2018).

**Institutional Review Board Statement:** Not applicable.

**Informed Consent Statement:** Not applicable.

**Data Availability Statement:** Not applicable.

**Conflicts of Interest:** The authors declare no conflict of interest.

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
