# Peer review of "Combustion and Emission Analysis of Spent Mushroom Compost and Forestry Woodchip for Management and Energy Production"

_fire, doi:10.3390/fire6010009_

Round 1

Reviewer 1 Report

Summary:

The paper studies combustion of spent mushroom compost and forestry woodchip in fluidized bed. The results have been obtained in a small scale (500 kWth) bubbling fluidized bed unit. The analysis methods were mostly accurately documented with some forgotten details. The results support the industry, which seeks to improve the usage of this type of biomass as a fuel. I recommend to publish the article after the following issues have been solved.

Major issues:

1) The bed temperature in a BFB furnace is typically 800-900°C (e.g. https://doi.org/10.1021/ef700650x, https://doi.org/10.1016/j.energy.2017.04.036), even in gasification conditions (e.g. https://doi.org/10.3303/CET2292065, https://doi.org/10.1016/j.biortech.2014.10.045). Thus, the selected bed temperature (<650°C) is very low and does not represent the typical conditions in BFB combustion. What was the reasoning for selecting such a low temperature? Or was this due to properties of the tested fuel and increasing the bed temperature was not possible? The higher bed temperature would have a significant effect on the results. The achieved results have value on their own, but it would be good to mention already in the abstract that the bed temperature was lower than in typical BFB combustion conditions. Then the reader can quickly decide, if the reported data is valid for their purpose.

2) Line 173, Chapter 2.3. The correct sampling of solids is one of the most important aspects of small and large scale combustion tests. Describe the sampling procedure of bed material and ashes.

3) The test procedure was not described in Chapter 2. For example, the duration of the tests, possible bottom ash removal during the test (or maybe the ash was just accumulating to bed during the test?) and possible make-up feed.

Minor issues:

1) The grammar should be checked. For example, the first and last sentence of the abstract should be revised. The text contains several sentences, which are either unclear or have grammatical issues (e.g. line numbers 60 – 62).  I recommed to use a professional language editing service.

2) Line 72: “produced 6”?

3) Line 99, Chapter 2.1. The analysis methods and standards have been reported in details, but the method for defining the volatile content for the proximate analysis is not mentioned.

4) Line 111: For example the acceptance testing of steam generators (DIN 1942) requires that the ash content is defined by DIN 51719, in which the sample is burned at 815°C. This method has been applied in many research papers as well (e.g. https://doi.org/10.1115/1.4002690,  https://doi.org/10.1021/ie200537m, https://doi.org/10.1016/j.biombioe.2018.08.014, https://doi.org/10.1016/j.biombioe.2018.08.014, https://doi.org/10.1021/ef4014604). What was the basis for selecting the low ashing temperature?

5) Line 118: the boiler model is mentioned, but a drawing or a photograph would provide a better understanding of the test equipment. For example, the location of secondary air injections? What were the dimensions of the equipment? How was the heat exchanger arranged (water cooled or actually evaporative or superheating, location)? What are the “two combustion chambers”, how they are connected?

6) Line 119: “boiling fluidized bed principle”. Remove “boiling”. It is just fluidized bed principle.

7) Line 123: “two combustion chambers”.

8) Line 139, Table 1. “Fumes flow”. Do you mean flue gas flow? Fumes is not a usually used term for this.

9) Consider integrating the Chapter 2.2.1 and Table 2 with the main chapter 2.2. There is no reason to keep this table separated in a sub-chapter.

10) Figure 1. In the review-version, the figure is not clear. Try to improve this for the final version.

11) Chapter 3.2.2. It is not fully clear, if the analyzed metal content in Table 5 is from the bottom ash or from the bed material. This is probably clarified after major issues 2 and 3 are solved. In any case, using “olivine” in the table title is misleading. Maybe “bed material” instead (after the test, the bed material is a mixture of olivine and ash from fuel).

12) The composition of the bed material or the enrichment of different metals depends on the share of ash in the bed after the test. Probably the test time has not been long enough to reach steady-state (or maybe there has not been constant bottom ash removal during the test – just accumulating the ash to the bed) and the result depends on the test time (increasing enrichment as a function of time). Can these results be applied to evaluate long term combustion (i.e. weeks or months)?

13) A further note related to above: in a large scale BFB, the quality of the bed material is constantly controlled by removing the bottom ash (i.e. the enriched material) and feeding new make-up material (e.g. olivine). I assume that in this test, there was no make-up feed during the test – please confirm.

14) Line 280: were the gas emissions stable during the test? Or perhaps fluctuating or increasing?

15) Line 316: do you mean fly ash, i.e. the ash collected by the multi-cyclone and bag filter (combined)?

16) Line 365: the high oxygen value is not an indication of good combustion conditions. The oxygen after the combustion is determined by the selected excess air coefficient (or air-to-fuel -ratio). On line 287, the oxygen has been mentioned as about 15%, which is quite high and indicates a poor efficiency (too high air/fuel-ratio). Typical oxygen concentration in a commercial BFB boiler is about 5%.

17) What is the main contribution of the article (besides the need for further tests)? You need to clarify this better in the conclusions. Did you reach your objectives?

18) The references include some very old references. Consider, if they are really necessary.

Author Response

Summary:
The paper studies combustion of spent mushroom compost and forestry woodchip in fluidized bed. The results have been obtained in a small scale (500 kWth) bubbling fluidized bed unit. The analysis methods were mostly accurately documented with some forgotten details. The results support the industry, which seeks to improve the usage of this type of biomass as a fuel. I recommend to publish the article after the following issues have been solved.
Major issues:
1) The bed temperature in a BFB furnace is typically 800-900°C (e.g. https://doi.org/10.1021/ef700650x, https://doi.org/10.1016/j.energy.2017.04.036), even in gasification conditions (e.g. https://doi.org/10.3303/CET2292065, https://doi.org/10.1016/j.biortech.2014.10.045). Thus, the selected bed temperature (<650°C) is very low and does not represent the typical conditions in BFB combustion. What was the reasoning for selecting such a low temperature? Or was this due to properties of the tested fuel and increasing the bed temperature was not possible? The higher bed temperature would have a significant effect on the results. The achieved results have value on their own, but it would be good to mention already in the abstract that the bed temperature was lower than in typical BFB combustion conditions. Then the reader can quickly decide, if the reported data is valid for their purpose.

Dear Reviewer, thank you for your comments. Examples of fluidized bed combustion are reported in the literature, with even lower bed temperatures (e.g. 600°C in https://doi.org/10.1016/j.fuel.2022.127007 ). Therefore the temperature used is in line with other similar works.

2) Line 173, Chapter 2.3. The correct sampling of solids is one of the most important aspects of small and large scale combustion tests. Describe the sampling procedure of bed material and ashes.

Thank you for the comment. We add the following description in the text (Chapter 2.3.): During the process, bed material (composed by olivine/sand and bottom ashes) is extracted continuously by mean a discharge system, which provide the separation of the bottom ashes from the olivine/sand and the re-feeding of the olivine/sand into the bed. Sampling was conducted by extracting bed material samples from the discharge system before to the re-feeding phase.

3) The test procedure was not described in Chapter 2. For example, the duration of the tests, possible bottom ash removal during the test (or maybe the ash was just accumulating to bed during the test?) and possible make-up feed.

Thank you for the comment. We add the following description in the text (Chapter 2): The power plant operating in continuous mode, regulating the power output as request by the user; plant switch off is realized for maintenance at scheduled intervals. Test duration was about six hours, during which the plant power output has been maintained as much as possible in a steady way.

Minor issues:
1) The grammar should be checked. For example, the first and last sentence of the abstract should be revised. The text contains several sentences, which are either unclear or have grammatical issues (e.g. line numbers 60 – 62). I recommed to use a professional language editing service.

Thanks for the comment, we conducted an extensive english check.

2) Line 72: “produced 6”?

Thank you for your comment. The text has been corrected.

3) Line 99, Chapter 2.1. The analysis methods and standards have been reported in details, but the method for defining the volatile content for the proximate analysis is not mentioned.

Thank you for your comment. As required we have specified that was uesd the method as reported in a cited article.

4) Line 111: For example the acceptance testing of steam generators (DIN 1942) requires that the ash content is defined by DIN 51719, in which the sample is burned at 815°C. This method has been applied in many research papers as well (e.g. https://doi.org/10.1115/1.4002690, https://doi.org/10.1021/ie200537m, https://doi.org/10.1016/j.biombioe.2018.08.014, https://doi.org/10.1016/j.biombioe.2018.08.014, https://doi.org/10.1021/ef4014604). What was the basis for selecting the low ashing temperature?

Thank you for your comment. As written in the text, the methods used is that described in ISO 18122:2016 “Solid Biofuels - Determination of ash content”

5) Line 118: the boiler model is mentioned, but a drawing or a photograph would provide a better understanding of the test equipment. For example, the location of secondary air injections? What were the dimensions of the equipment? How was the heat exchanger arranged (water cooled or actually evaporative or superheating, location)? What are the “two combustion chambers”, how they are connected?

Thank you for the comment. Figure 1 is now add in the paper. As you can see, the boiler drawing simplified scheme; the heat exchanger is positioned downstream the oxidating chamber and produce approximately 500 kWt. Thermal power is provided to the user by mean superheating water. The two combustion chambers (accelerating and stationary flue gas) are connected as represented in the Figure 1. The boiler, the brand and the operation system are currently under patent application and therefore producers do not want to publish more photos/information.

6) Line 119: “boiling fluidized bed principle”. Remove “boiling”. It is just fluidized bed principle.

Thank you for comment, the text has been corrected.

7) Line 123: “two combustion chambers”.

Thank you for comment. See the above mentioned description at point 5.

8) Line 139, Table 1. “Fumes flow”. Do you mean flue gas flow? Fumes is not a usually used term for this.

Thank you for the comment. Gas flow is now used instead of Fumes flow

9) Consider integrating the Chapter 2.2.1 and Table 2 with the main chapter 2.2. There is no reason to keep this table separated in a sub-chapter.

Thanks for the comment. great suggestion, the sub-paragraph has been deleted.

10) Figure 1. In the review-version, the figure is not clear. Try to improve this for the final version.

Thank you for the comment. The figure has been enlarged. it is in excellent resolution, so it can be reformatted according to the needs of the journal.

11) Chapter 3.2.2. It is not fully clear, if the analyzed metal content in Table 5 is from the bottom ash or from the bed material. This is probably clarified after major issues 2 and 3 are solved. In any case, using “olivine” in the table title is misleading. Maybe “bed material” instead (after the test, the bed material is a mixture of olivine and ash from fuel).

Thank you for the comment. was chosen to change the headings in the table with "olivina" and "bed material". The first represents the minieral as such, while the second is the final mixture in the combustion chamber that contains olivine and bottom ash.

12) The composition of the bed material or the enrichment of different metals depends on the share of ash in the bed after the test. Probably the test time has not been long enough to reach steady-state (or maybe
there has not been constant bottom ash removal during the test – just accumulating the ash to the bed) and the result depends on the test time (increasing enrichment as a function of time). Can these results be applied to evaluate long term combustion (i.e. weeks or months)?

Thank you for comment. After answering point 2 and 3 of the major issues, we hope that this concept is clarified.

13) A further note related to above: in a large scale BFB, the quality of the bed material is constantly controlled by removing the bottom ash (i.e. the enriched material) and feeding new make-up material (e.g. olivine). I assume that in this test, there was no make-up feed during the test – please confirm.

Thank you for comment. After answering point 2 and 3 of the major issues, we hope that this concept is clarified

14) Line 280: were the gas emissions stable during the test? Or perhaps fluctuating or increasing?

Thank you for comment. The graphs of the emissive trend of the atmospheric macropollutants have been added

15) Line 316: do you mean fly ash, i.e. the ash collected by the multi-cyclone and bag filter (combined)?

Yes, by fly ash we mean those that are stopped by abatement systems. The fly ash sampling has been performed at the end of the test, taking the ash champion into the tank positioned downstream the filtering section of the plant (multi-cyclone and bag filter combined)

16) Line 365: the high oxygen value is not an indication of good combustion conditions. The oxygen after the combustion is determined by the selected excess air coefficient (or air-to-fuel -ratio). On line 287, the oxygen has been mentioned as about 15%, which is quite high and indicates a poor efficiency (too high air/fuel-ratio). Typical oxygen concentration in a commercial BFB boiler is about 5%.

Thanks for the comment. Actually the concentration of O2 is not very low. However, what we wanted to say is that the low CO concentration indicates a combustion in good stoichiometric conditions and this therefore suggests that there is a low profile of organic pollutants in emission (which are produced in large quantities under partial and incomplete biomass combustion conditions). The text has been amended to clarify the concept.

17) What is the main contribution of the article (besides the need for further tests)? You need to clarify this better in the conclusions. Did you reach your objectives?

Thanks for the comment, the conclusions have been corrected.

18) The references include some very old references. Consider, if they are really necessary.

Thanks for the annotation. We have checked and deleted older references that weren't really necessary and added recent references.

Reviewer 2 Report

The manuscript "Combustion and emission analysis of spent mushroom  compost and forestry woodchip for management and energy production" by Carnevale et al. considers a specific biomass mixture for energy production by combustion. Although very important research nowadays the impact of the present conclusions are limited sometimes (too) obvious and will have minor impact. The goal of the research is not very adequately described and it looks more like a measurement report. Furthermore see many spelling errors as well as unclear and puzzling formulations with sentences that are too long and unclear assembled. To start already at the abstract, e.g.:

1) Abetment

2) SO2 values appear (with?) respect (to?) the ...

3) therefore is stictly required

and in the conclusions:

1) the appearance of "in fact" in the second sentene at that place.

2) "...it emerged the need..."

3) "2.5 fraction"

The abstract and conclusion should be written in order to be understandable and independent from the remaining body!

The structure of the remainder should be partitioned in paragraphs and more issues are detected.

First, start to make it readable and understandable as well!!

Author Response

The manuscript "Combustion and emission analysis of spent mushroom compost and forestry woodchip for management and energy production" by Carnevale et al. considers a specific biomass mixture for energy production by combustion. Although very important research nowadays the impact of the present conclusions are limited sometimes (too) obvious and will have minor impact. The goal of the research is not very adequately described and it looks more like a measurement report. Furthermore see many spelling errors as well as unclear and puzzling formulations with sentences that are too long and unclear assembled. To start already at the abstract, e.g.:
1) Abetment
2) SO2 values appear (with?) respect (to?) the ...
3) therefore is stictly required
and in the conclusions:
1) the appearance of "in fact" in the second sentene at that place.
2) "...it emerged the need..."
3) "2.5 fraction"
The abstract and conclusion should be written in order to be understandable and independent from the remaining body!
The structure of the remainder should be partitioned in paragraphs and more issues are detected.
First, start to make it readable and understandable as well!!

Dear Reviewer, thank you for reviewing and commenting. We greatly appreciated your annotations and proceeded to make all the corrections and modifications requested.

1) Abetment Corrected in the text

2) SO2 values appear (with?) respect (to?) the … Changed in the text

3) therefore is strictly required Changed in the text

and in the conclusions:

1) the appearance of "in fact" in the second sentene at that place. Changed in the text

2) "...it emerged the need..." Changed in the text

3) "2.5 fraction" Changed in the text
The abstract and conclusion should be written in order to be understandable and independent from the remaining body!

"The structure of the remainder should be partitioned in paragraphs and more issues are detected.
First, start to make it readable and understandable as well!!"

Dear Reviewer, Thank you for your valuable comments. We modified the text of the abstract and conclusion as required. Moreover, as regards the remaining text structure, we have made the recommended changes.

Reviewer 3 Report

This is a high-quality manuscript. It studies the combustion and emission of spent mushroom compost and forestry woodchip. This topic is of interest due to the popularity and importance of biomass.

The introduction and background are sufficient. The motivation is clear, and all the relevant references are included. 

The configuration of the experimental setup is introduced with enough details. 

In the result and discussion section, the author provides reasonable explanations of each experimental observation. 

I only have some minor comments.

1. In section 3.3, the "excellent combustion conditions" seems a little confusing to me. What does this refer to? Does this mean "complete combustion"?
2. Some figures have low quality. The author should provide a high-quality version of these figures. For example, in figure 1, the texts are too small to see.

The manuscript is satisfactory, and I would recommend it publish on Fire once these minor comments are addressed in the manuscript.

Author Response

This is a high-quality manuscript. It studies the combustion and emission of spent mushroom compost and forestry woodchip. This topic is of interest due to the popularity and importance of biomass.
The introduction and background are sufficient. The motivation is clear, and all the relevant references are included.
The configuration of the experimental setup is introduced with enough details.
In the result and discussion section, the author provides reasonable explanations of each experimental observation.
I only have some minor comments.

Dear Reviewer Thank you for the motivating and encouraging comments. Thank you for taking the time to read the manuscript and for recommending improvements. As requested, we made changes to the text, specifying and clarifying the highlighted points.

1. In section 3.3, the "excellent combustion conditions" seems a little confusing to me. What does this refer to? Does this mean "complete combustion"?

Thank you for the comment. We have replaced the term “excellent” which could be misunderstood with “good”. By optimal combustion conditions we mean that the process conditions under which the combustion took place were optimal for the purposes of producing low concentrations of emissions into the atmosphere. Combustion cannot be defined as complete because it would be an ideal condition, difficult to reach in experimental test.

2. Some figures have low quality. The author should provide a high-quality version of these figures. For example, in figure 1, the texts are too small to see.

Thank you for your comment. We have checked the resolution of the figures and specifically enlarged the Figure in order to make the text clearly readable.

The manuscript is satisfactory, and I would recommend it publish on Fire once these minor comments are addressed in the manuscript.

Thank you for appreciating our work, this gives us motivation to continue the research

Round 2

Reviewer 1 Report

Some of the issues were not solved. The numbers refer to the issue numbers of the original review.

Major issues:

1) In the referred work (https://doi.org/10.1016/j.fuel.2022.127007) the bed temperature was 600°C only at the start of the fuel feeding. In the actual test conditions, the bed temperature was 850°C (“This work sets the Tb in all test conditions at 850 °C”). So, this article further proves the point that the typical bed temperature in BFB combustion is 800-900°C. This issue has not been solved and your added reference to the above article is false.

Minor issues:

1) Check English again. Some examples:

- Despite biomass energetic exploitation represents a sustainable practice and contributes to produce clean energy, biomass burning emissions, due to their high particulate matter concentration, are recognized among the major global environmental risk factors.
- Thermal power is provided to the user by mean superheating water.
- In the LASER-B (Laboratory of Experimental Renewable Energies from Biomass) of CREA-IT were conducted the analysis of physical and chemical parameters of the materials supplied by an historical mushroom farm located in Lazio, Italy, and where is placed the biomass test plant realized by Aster Energetica Srl.
- The olivine before and 234 after the process and the ash were also mineralized (?) to evaluate the metal content in ICP-235 MS.
- Regards SMC properties is important to underline that the compost is a highly volatile matter biomass given the presence of straw; the fixed carbon is very low, therefore the combustion that takes place in the bed is the minor part, the bulk of the combustion takes place in the downstream chambers and obviously before the exchangers where all the volatile part is oxidized.

3) You should simply refer to the appropriate standard instead of a conference article. For example, DIN 51720 is one such standard, which describes the definition of volatiles. You may have used a different standard – consult your laboratory personnel.

4) The question was not answered. Please, explain why you selected the low ashing temperature.

16) In the first version, which I reviewed, the text was as follows:

during the sampling, high O2 values and low CO values were recorded”.

which was the reason for my comment.

It seems that at some point the text has been changed to “low O2 values”, which of course makes more sense.

I do not know, why the text was reading “high O2” in my version, but in any case, this issue is now solved.

Author Response

Some of the issues were not solved. The numbers refer to the issue numbers of the original review. 

Major issues: 

1) In the referred work (https://doi.org/10.1016/j.fuel.2022.127007) the bed temperature was 600°C only at the start of the fuel feeding. In the actual test conditions, the bed temperature was 850°C (“This work sets the Tb in all test conditions at 850 °C”). So, this article further proves the point that the typical bed temperature in BFB combustion is 800-900°C. This issue has not been solved and your added reference to the above article is false. 

Thanks for the comment. The previously added citation has been removed, thank you for reporting the error. The biomass considered has an elevated moisture content, which inevitably reduces temperature of bed and the efficiency of the thermochemical conversion process. This mixture leads to a bed temperature of about 650°C. Despite is a lower temperature compared to other similar works in the literature, it has been observed that during the operative parameters optimization phase, this temperature allows to keep the bed in stable temperature condition and subsequently led to have correct oxidation in the combustion chambers. Moreover, the lower temperature also reduces the production of NOx in emission. For completeness, this concept has also been expressed in the paper. 

Minor issues: 

1) Check English again. Some examples: 

- Despite biomass energetic exploitation represents a sustainable practice and contributes to produce clean energy, biomass burning emissions, due to their high particulate matter concentration, are recognized among the major global environmental risk factors. 
- Thermal power is provided to the user by mean superheating water. 
- In the LASER-B (Laboratory of Experimental Renewable Energies from Biomass) of CREA-IT were conducted the analysis of physical and chemical parameters of the materials supplied by an historical mushroom farm located in Lazio, Italy, and where is placed the biomass test plant realized by Aster Energetica Srl. 
- The olivine before and 234 after the process and the ash were also mineralized (?) to evaluate the metal content in ICP-235 MS. 
- Regards SMC properties is important to underline that the compost is a highly volatile matter biomass given the presence of straw; the fixed carbon is very low, therefore the combustion that takes place in the bed is the minor part, the bulk of the combustion takes place in the downstream chambers and obviously before the exchangers where all the volatile part is oxidized. 

Thank you. We checked English again 

3) You should simply refer to the appropriate standard instead of a conference article. For example, DIN 51720 is one such standard, which describes the definition of volatiles. You may have used a different standard – consult your laboratory personnel. 

We added the method used. 

4) The question was not answered. Please, explain why you selected the low ashing temperature. 

We used a different standard then the one you mentioned (DIN 51719). The standard we use is the UNI EN ISO 18122:2016 which implements the European standard ISO 18122:2015 recently further revised by ISO 18122:2022. Here the solid fuel ash determination is made using the two temperature ramps described in the paper. We always try to use the most up-to-date standards possible. 

16) In the first version, which I reviewed, the text was as follows: 

during the sampling, high O2 values and low CO values were recorded”. 

which was the reason for my comment. 

It seems that at some point the text has been changed to “low O2 values”, which of course makes more sense. 

I do not know, why the text was reading “high O2” in my version, but in any case, this issue is now solved. 

Maybe we corrected it without marking the review. "high O2" was a typo.